# Unlocking cinnamon export success: Key determinants from the world's top five producers

**Krishantha Wisenthige**[1]*, **Ruwan Jayathilaka**[2], **Umesha Dabare**[1],
**Thisalya Marasinghe**[1‡], **Malki Radeesha**[1‡], **Fiona Ann**[2‡], **Nethmi Kavindya**[2‡]

**1** Department of Business Management, Sri Lanka Institute of Information Technology, Malabe, Sri Lanka,
**2** Department of Information Management, Sri Lanka Institute of Information Technology, Malabe, Sri Lanka

☯ These authors contributed equally to this work.
‡ TM, MR, FA and NK also contributed equally to this work.
* krishantha.w@sliit.lk

## Abstract

The purpose of this research study is to identify the factors affecting cinnamon export income (CEI) in the main five cinnamon export countries, namely China, Sri Lanka, Indonesia, Madagascar and Vietnam for the period from 1992–2022. Secondary data was sourced from the Food and Agriculture Organization and World Bank. Based on the past literature, it has been found out that production volume (PV), domestic consumption (DC), exchange rate (ER) and cultivated land area (CLA) significantly impact on CEI. Simple Linear Regression models were applied to analyse the impact of the identified factors affecting CEI in the present study. The findings revealed, PV negatively impacts the export income of cinnamon in China, Sri Lanka, and Vietnam, while having a positive impact on Indonesia and Madagascar. Moreover, while DC appears to have a positive impact in Sri Lanka, it has a negative impact in China, Vietnam, Indonesia and Madagascar for the same. Accordingly, ER is positive for countries Madagascar, Sri Lanka, and Vietnam while adverse for Indonesia and China. In addition, the study proved that CLA positively influences CEI of China, Vietnam, and Madagascar but negatively for Sri Lanka and Indonesia. Consequently, the findings from this study greatly assist policymakers, exporters, and the industry professionals in executing strategies to enhance the export income & thereof export practices of cinnamon. Finally, this research addresses several gaps in cinnamon export studies, supporting sustainable growth and competitiveness in the sector.

## 1. Introduction

Cinnamon, whose origin is in Sri Lanka and the Southern Indian Malabar Coast, was the first among spices to reach the Mediterranean and is a product derived from the bark of evergreen tropical trees in the genus *Cinnamomum* [1]. Even though there are around 250 varieties, a few are commercially grown, namely Ceylon and Cassia

**Data availability statement:** The data collected from secondary data sources such as the Food and Agriculture Organization and World Bank data and data used for the study available in S1 Appendix.

**Funding:** The author(s) received no specific funding for this work.

**Competing interests:** The authors have declared that no competing interests exist.

cinnamon, which are most dominant in production and exportation worldwide [2]. The global need for cinnamon continues to exceed supply, as a result of its recognized therapeutic potential in the treatment of numerous human conditions. It is widespread in application in facilitating weight loss, bringing down blood sugar and cholesterol levels, and suppressing cardiovascular risks [3,4]. Studies have proved that cinnamon regulates blood glucose levels by minimizing glucose spikes [5,6]. Additionally, cinnamon powder enhances immune system function and general physiological well-being [7]. Daily intake of 1500 mg has been proved effective in curbing Non-Alcoholic Fatty Liver Disease (NAFLD) by decreasing liver fat, increasing liver function, and inflammation reduction; it was also used during the COVID-19 outbreak in relieving respiratory issues, cough, and accumulation of mucus [8,9].

The main producers of True Cinnamon (*Cinnamomum zeylanicum*) are Madagascar and Sri Lanka, with Sri Lanka providing close to 90% of world supply [1,10], and cassia cinnamon, comprising more than 90% of traded cinnamon, being mainly exported by China, Vietnam, Madagascar, and Indonesia [11]. The paper attempts to explore export income determinants in cinnamon by targeting the world's first five largest cinnamon-exporting countries. Previous work on Indonesia points out that the price of cinnamon, the exchange rate, and export volume are positively significant in impacting its export of cinnamon [12]. Large-scale agronomic works confirmed that export income is significantly affected by attributes including production volume (PV), cultivated land area (CLA), exchange rate (ER), and domestic consumption (DC) [13–16]. The majority of cinnamon work, however, remains as single-country case studies, mainly on Sri Lanka and Indonesia, and without a comparative worldwide dimension [17,18]. This work fills that vacuum by considering export performance in the world's first five most important countries and attempting to increase their competitiveness in the worldwide cinnamon business [19]. Accordingly, the main aim of this research study is to examine the factors that influence cinnamon export income (CEI) in the top five countries that export cinnamon (China, Sri Lanka, Vietnam, Indonesia, and Madagascar). Conversely, the secondary aim of this research study is to examine the factors that influence CEI in each of the countries highlighted above. This research study will primarily help to address the gap found in previous studies and offer important information that can help in making appropriate decisions in the international cinnamon market. The contribution of productions to export earnings is multidimensional in that previous studies clarify that production level and quality are key in improving export performance. This paper discusses how differentials in the volume of production among key cinnamon-exporting nations shape export income, while the land area also determines export income. Additionally, the dynamics of the exchange rate are complex, while depreciation increases price competitiveness, it also generates volatility. DC is a significant factor in determining the availability of goods for export because consumption creates a balancing act. Accordingly, the significance of this study can be highlighted in several ways. Firstly, this study provides critical insights into the varying impacts of PV on CEI across the world's top five producers. In second, the findings on DC elaborate wide opportunities of implementing differentiated marketing and production approaches, as DC positively influences CEI

in Sri Lanka but negatively in other major producers. Based upon the findings of the studies, thirdly, it is stated that the fluctuations in ER provide valuable information in financial planning, indicating that the changes in the value of currencies influence cinnamon export incomes in various ways in different countries, thereby necessitating the formulation of specific foreign exchange risk management strategies. Furthermore, upon scrutinizing the importance of CLA in export incomes of cinnamon, this analysis will help in optimizing land allocation strategies to improve export performance in various regions. Lastly, at the end of this analysis, the findings of this research, which are specific to the countries, will provide valuable information to international trade practitioners, investors, as well as countries, in formulating effective sustainable export strategies in the international markets, ensuring long-term viability in the international markets. This analysis, despite dealing with one type of spices only, will be of great utility in making financial decisions of countries, since the conclusion of this analysis study can be replaced with that of export of spices.

## 2. Methods

### 2.1. Data sources

The sources of the secondary data collected for this research are presented in Table 1, and this study focuses on analysing the factors affecting cinnamon export in China, Sri Lanka, Vietnam, Indonesia, and Madagascar.

Table 1 covers the data sources of this study, and it uses the period of 1992–2022, retrieved from reliable databases such as the FAOSTAT and the World Bank. Data used in the study is fully available in S1 Appendix. The variables which would be put into consideration in this study are CEI, PV, DC, ER, and CLA. CEI, is the return on exports of cinnamon and is measured in USD. Cinnamon PV is taken as a percentage of GDP per capita, while it is measured in tones DC is a variable measured in Kg, showing the sum of cinnamon consumed within the country. CLA is expressed in hectares (ha) per capita. ER is measured in USD.

### 2.2. Methodology

In order to structure this research in the form of evidence-based practice, a conceptual framework was developed, as indicated in Fig 1, stating that CEI is a result of four important determinants: PV, DC, ER, and CLA. These variables were chosen based on prominent literary works that emphasize the importance of these factors in agricultural as well as export countries. It has been observed in various empirical studies that increased production outcomes result in increasing exportable surpluses with higher foreign revenues, mainly in the form of commodities such as tea, sugar, and apples

**Table 1. Data and sources of the parameters.**

| Variable | Measure | Sources |
|---|---|---|
| CEI | USD (% of Total export income) | Food and Agricultural Organization of the United Nations (FAOSTAT) https://www.fao.org/faostat/en/#data/QCL |
| PV | Tons (% of per capita GDP) | Food and Agricultural Organization of the United Nations (FAOSTAT) https://www.fao.org/faostat/en/#data/QCL |
| DC | Kg (per capita) | Food and Agricultural Organization of the United Nations (FAOSTAT) https://www.fao.org/faostat/en/#data/QCL |
| CLA | Hectare (% of total land area) | Food and Agricultural Organization of the United Nations (FAOSTAT) https://www.fao.org/faostat/en/#data/QCL |
| ER | USD (LCU per US$, Period Average) | The World Bank https://wdi.worldbank.org/table/4.16 https://data.worldbank.org/indicator/PA.NUS.FCRF?locations=LK https://data.worldbank.org/indicator/PA.NUS.FCRF |

Source: Authors' compilation based on survey data collected.

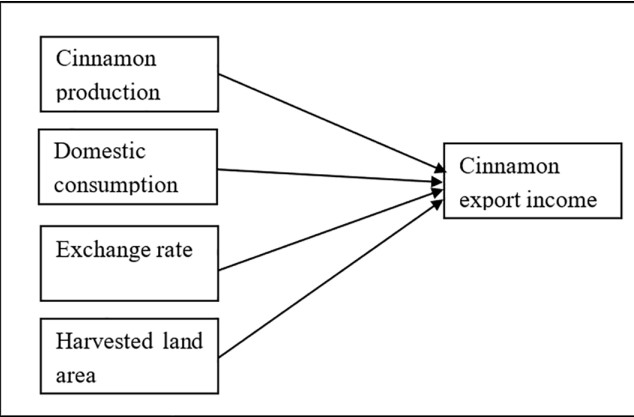

**Fig 1. Conceptual model of the variables.**

[20–23]. CLA has been identified as a factor in increasing export surpluses in terms of increased usages of resources with economies of scale in countries like Sweden, Indonesia, and Southeast Asia [24–27]. ER is identified as a determinant that affects export outcomes in terms of pricing tenability, whereupon a fall in the value of the money results in increased exports, yet inconsistent fluctuations result in severe outcomes [28–32]. At the same time, however, DC is identified as a factor that either drains away commodities towards foreign countries or, when restrained, results in increased exports by businesses [33–37].

The authors of this study have focused on Simple Linear Regression (SLR) models in conducting the analysis. The time period of 30 years was divided into 3 periods to analyse if CEI, PV, DC, ER and CLA has increased compared to the previous years in the recent years. Here, the average values of the variables were taken into consideration. Bubble maps, line graphs show how the identified variables can have an impact on CEI. The authors have A SLR model was used in this analysis to quantify the effects of PV, DC, CLA, and ER on CEI in the five major countries that export cinnamon. This model is essential in understanding the effects of those variables on the outcomes of the economic issues relating to the export of cinnamon between the years 1992 and 2022.

Equation 1 was used to conduct the SLR for variables PV, DC, CLA and ER for each country. The SLR was used to analyse the impact of the independent variables to the dependent variable CEI.

$$Y_{it} = \beta_0 + \beta_1 X_{it} + \varepsilon_{it} \tag{1}$$

Equation 2 was used to conduct the SLR and analyse the impact of PV on CEI.

$$CEI_{it} = \beta_0 + \beta_1 PV_{it} + \varepsilon_{it} \tag{2}$$

Equation 3 was used to conduct the SLR and to analyse the impact of CLA on CEI.

$$CEI_{it} = \beta_0 + \beta_1 HLA_{it} + \varepsilon_{it}. \tag{3}$$

Equation 4 was used to conduct the SLR and analyse the impact of DC on CEI.

$$CEI_{it} = \beta_0 + \beta_1 DC_{it} + \varepsilon_{it} \tag{4}$$

Equation 5 was used to conduct the SLR and analyse the impact of ER on CEI.

$$CEI_{it} = \beta_0 + \beta_1 ER_{it} + \varepsilon_{it} \tag{5}$$

While the study employs SLR models for each country to estimate the relationship between CEI and its potential determinants (PV, DC, CLA, ER), this approach was selected to ensure interpretability and manage country-specific heterogeneity. However, we acknowledge that SLR may not capture possible nonlinearities, interactions, or cross-country dynamics. Due to limitations in data uniformity across the full sample, advanced techniques such as panel regression or structural equation modelling were deemed infeasible within this study's scope. These limitations are discussed under the recommendations for future research.

## 2.3. Ethical consideration

This study is based exclusively on secondary data obtained from publicly available sources. The dataset contains no personal identifying information, and no human participants were directly involved. Therefore formal ethical approval and informed consent were not required.

## 3. Results and discussion

Secondary data was used in this research study, and a detailed overview of the analysis is discussed in this section. Moreover, the authors of this study have used the average values of the variables between the period of 1992–2022, SLR was conducted to analyse the impact of the independent variables to the dependant variable and the variations of the variables within the 30 years is illustrated by the scatter plots. The results generated for all the objectives are described clearly for the readers to get a clear understanding on this research. The overall aims of the study were to analyse the impact of DC, PV, CLA and ER to CEI in the main five cinnamon exporting countries in the world.

The following figures indicates the fluctuations of CEI, PV, DC, ER and CLA in the period of 30 years. Figs 2 and 3 show the analysis of CEI across the five main exporting countries brings into light the dominant position of Sri Lanka in the global cinnamon market. Sri Lanka has consistently reported the highest CEI value throughout the period from 1992 to 2022, recording a significant increase of 6.4402 USD in the latter period (2013–2022). This is attributed to many factors including the trading relations Sri Lanka has developed throughout history, the superior quality of the Ceylon cinnamon it has, and successful promotional campaigns which promote the product in international markets. Most importantly,

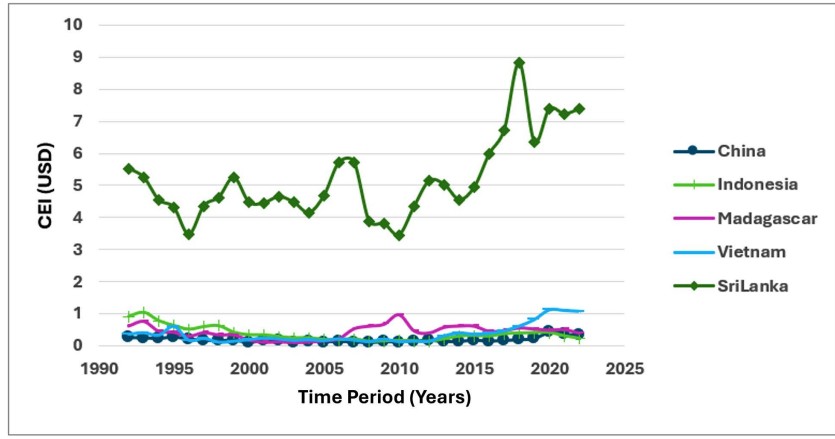

**Fig 2. Fluctuations of CEI between 1992-2022.**

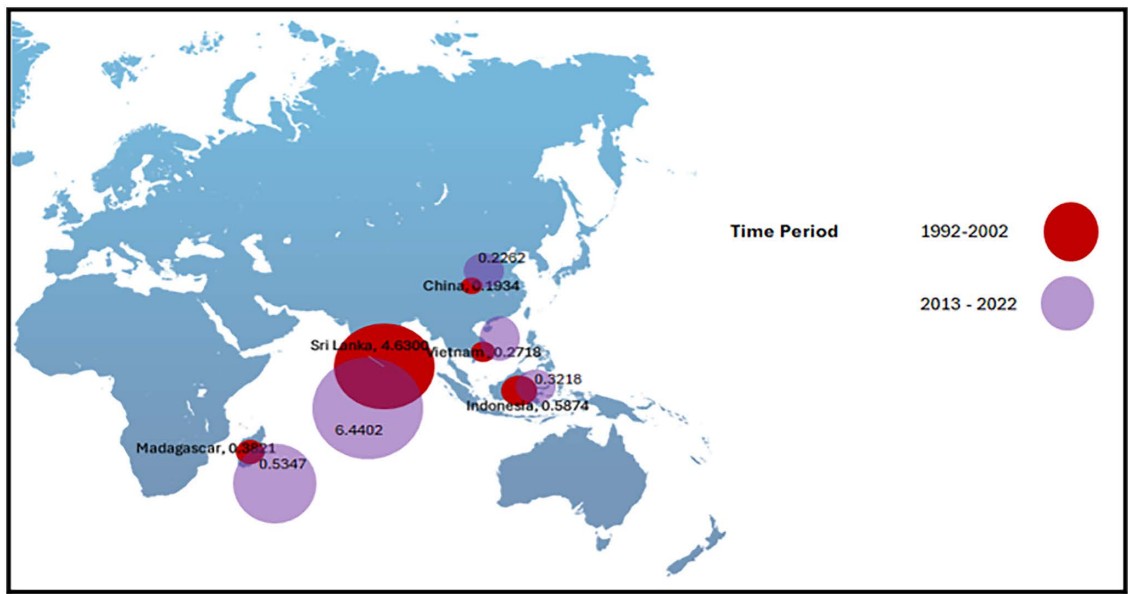

**Fig 3. Bubble Map for CEI.** Source.

improvements in quality can generate greater export receipts and hence enhance economic development for exporting nations. (CEI 1). Conversely, the impact of other countries involved in exporting activities, including China, Indonesia, Madagascar, and Vietnam, is that their export revenues have been flat or declined during this time. A particularly important factor is one that gives rise to a host of questions relating to a competitive approach that will ensure that Sri Lanka maintains this balance in the international arena. It is not only important that the continued steady rise of the CEI indicates that the export strategy of Sri Lanka is successful, but there is a certain important element that touches on the issue of quality enhancement in agricultural exports that will ensure international exposure of Indonesian businesses. (CEI 2).

Accordingly, Figs 4 and 5 show the variation of DC from 1992–2022. Variability in the DC of cinnamon is also high in the five major export countries, except in Sri Lanka, where this country has emerged as a notable outlier. This country shows a rather astonishing level of increased home consumption of cinnamon, peaking within the early years of the 2010s, indicating a rather stronger emphasis on home agricultural production in this country. This rather shows that this country has managed to develop a rather stronger home market in this commodity, in terms of rather stronger culture based as well as rather stronger economic factors that ensure rather stronger home use of this commodity. At the same time, the rather stronger home use of this commodity in Sri Lanka not only benefits the farmers within this particular country but also rather strengthens the overall export capacities of this particular country since this particular commodity is rather ensured a rather stronger home market. This dynamic has considerable implications for environmental sustainability, as increased local consumption may lead to more sustainable agricultural practices. Domestic consumption of such export crops in the richest countries presents huge implications not only for the home-based agricultural practices but also for the export capabilities and the environmental sustainability of the countries of origin [34,38].

Referring to Figs 6 and 7, those showcase the variation of ER. Analysis of ER in the five main exporting countries of cinnamon between the five countries suggests that there is a large gap between the periods of 1992–2002 and 2013–2022. This is indicated by the high ER of Vietnam in the latter period, showing that the value of Vietnam's money increased against the others. This is linked with the reduction in economic rents that will result in a positive change in the price that will benefit the poor in the rural as well as (ER 1). By contrast, the ER of China retreated, reflecting the broader

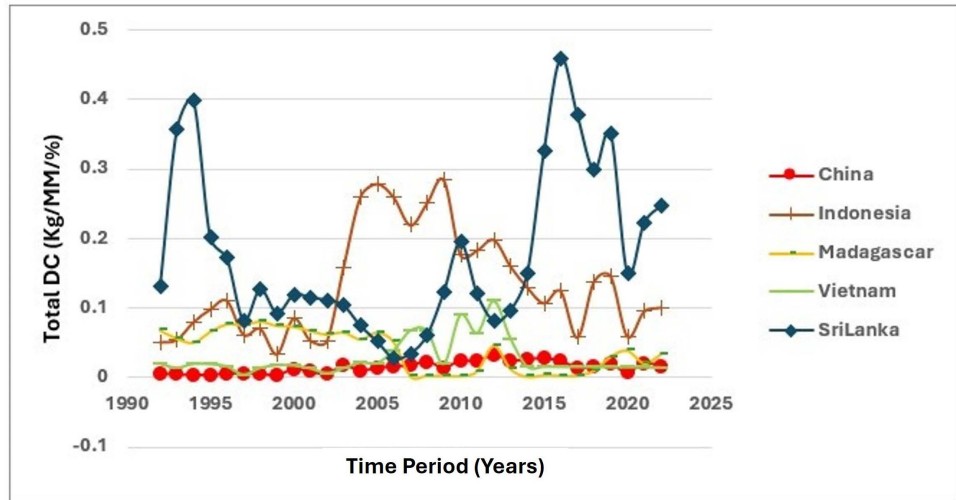

**Fig 4. Fluctuations of DC between 1992-2022.**

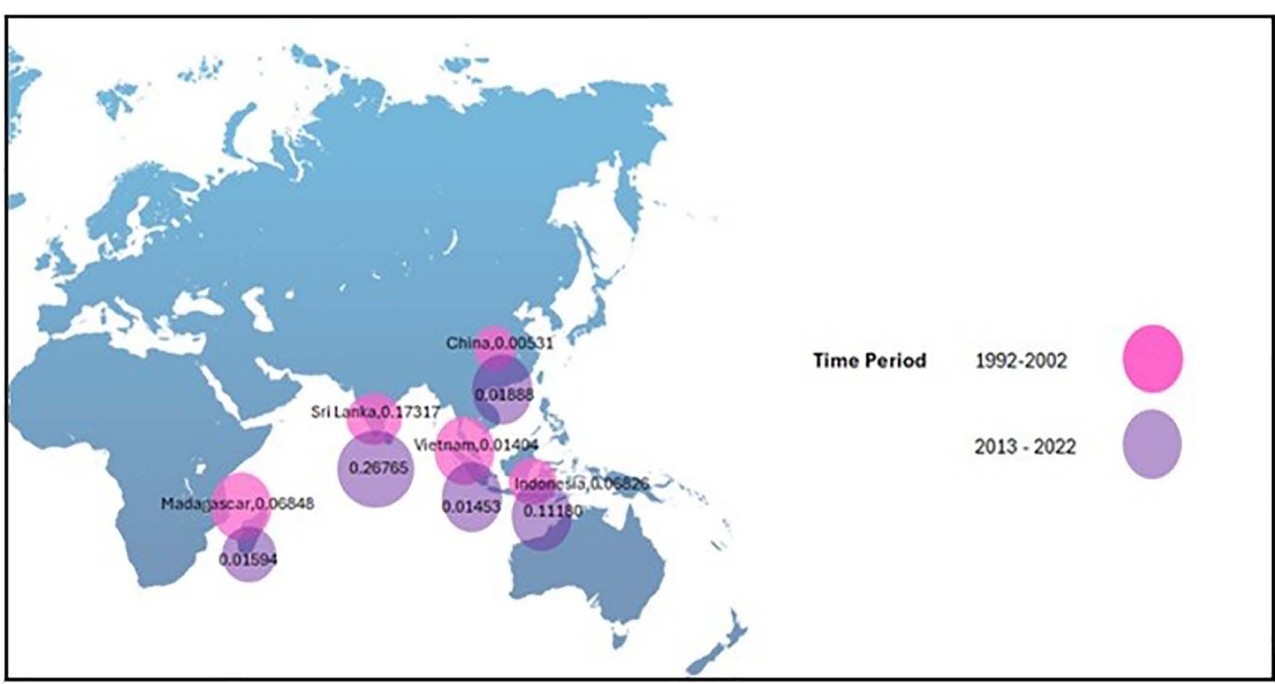

**Fig 5. Bubble map for CEI.**

economic challenges. The changes observed are caused by several factors, ranging from economic conditions to monetary policies, as well as changes occurring on the global market. A rise in the value of the Vietnamese currency will have an impact on the price of Vietnamese cinnamon on the global market, thus affecting its competitiveness on the export revenue. With the strength of the Vietnamese currency, the higher price of Vietnamese cinnamon would reduce its attractiveness to international buyers. Also, an appreciation of Brazil's real exchange rate erodes the competitive position of the

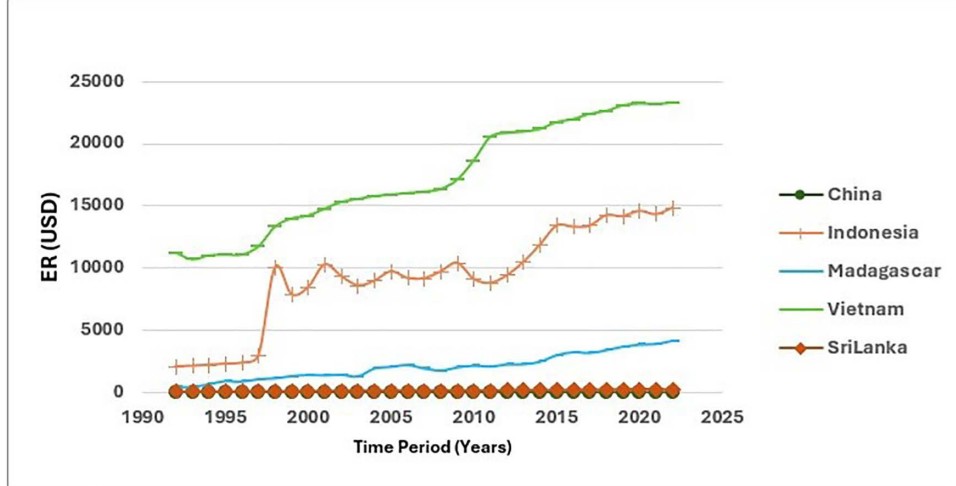

**Fig 6. Fluctuations of ER between 1992-2022.**

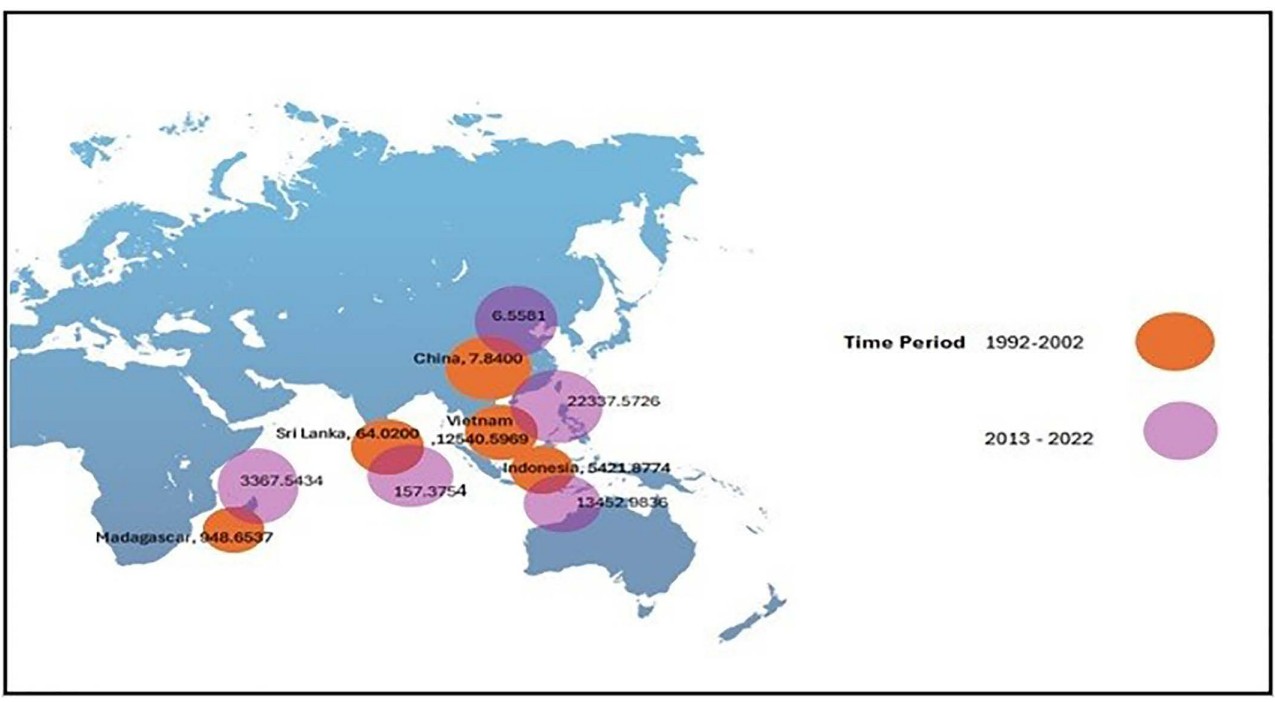

**Fig 7. Bubble map for CEI.**

sector within the country; this is further exacerbated by interest differentials, which are reflective of inherent weaknesses to be studied for possible improvement. (ER 2).

   The CLA analysis of cinnamon in the five major exporting countries shows important trends and differences according to Figs 8 and 9. Sri Lanka is considered to have the highest CLA in both periods analyzed-1992–2002 and 2013–2022-with a notable trend of increase in the last period. This growth underlines Sri Lanka's commitment to the expansion of its

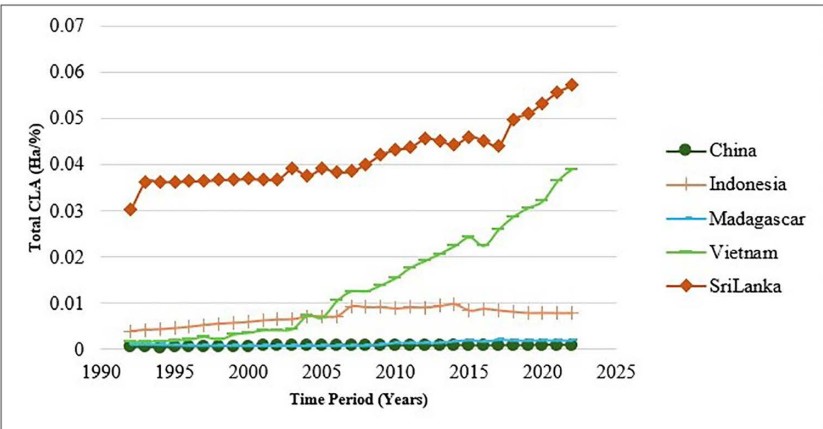

**Fig 8. Fluctuations of CLA between 1992-2022.**

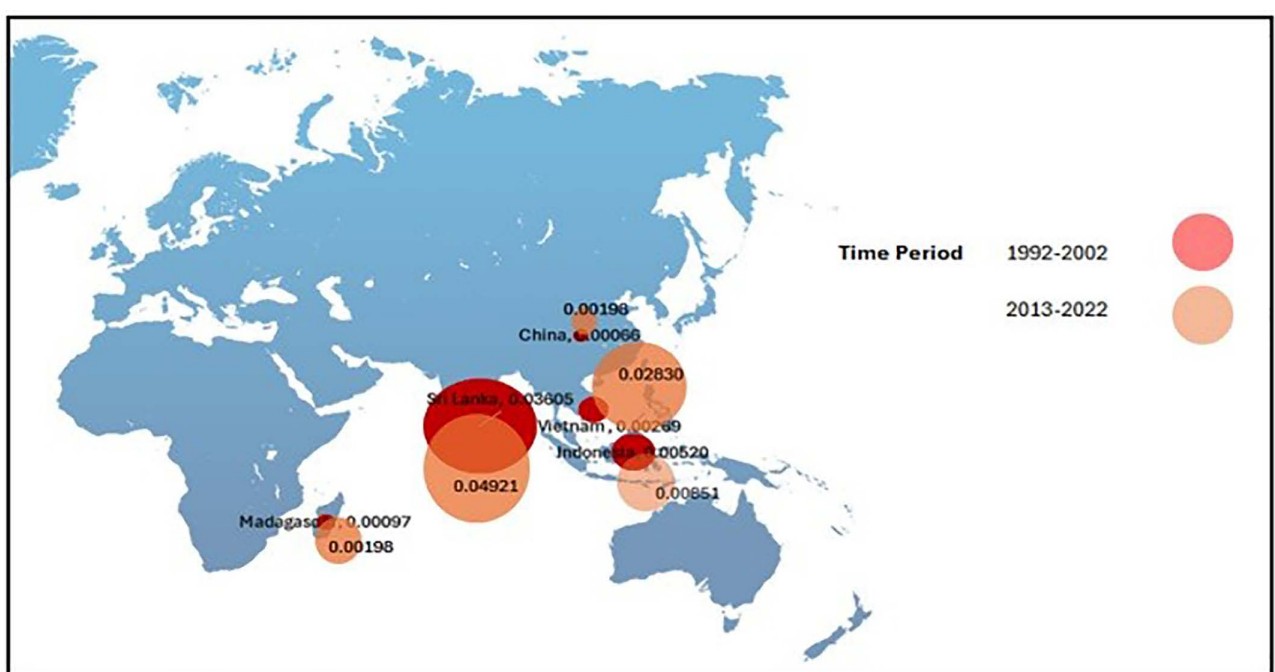

**Fig 9. Bubble map for CEI.**

cinnamon cultivation, in fact reflecting strategic investments into agricultural practices and land management. Vietnam also follows a more gradual increase in CLA, which reflects the continuously growing interest in cinnamon cultivation. Nutrients of the soil can affect the harvesting of crops and production [39]. Accordingly, China retains low land area for cinnamon farming, indicating the relatively lower role of that country in the global production of the spice. The large CLA of Sri Lanka and Vietnam reflects a strong commitment to enhancing their production capabilities, while the limited CLA of China indicates a minor focus on this production and trade within the context of changing agricultural landscapes.

The Figs 10 and 11 show the fluctuations of PV. At the same time, PV of cinnamon across the five main exporting countries presents a very interesting picture of the industry's dynamics over the periods 1992–2002 and 2013–2022. Indonesia and China showed the highest PV in both periods, indicating their large importance in global cinnamon production. But, over a worrying trend that develops as both nations show a decrease in PV in the latter period is observed; Indonesia especially suffered severely from the drastic drop in production in 2005. It is not something unique to Indonesia, however. Other countries also suffer decreases in cinnamon production, doubtless influenced by climate change and economic factors linked with GDP fluctuations. These changes in climatic conditions mostly affect the production of crops growing and its productivity [40] (PV 1). The relationship of the PV with the economic conditions indicates that the fluctuation of the GDP can seriously change the output of agriculture since usually in economic distress, a fall in production capacity takes place.

Accordingly, the following descriptive values were generated using the average values of the five variables CEI, DC, PV, CLA, and ER are shown in the Table 2 for the time frame of 1992–2002, 2003–2012, and 2013–2022 considering the countries, China, Sri Lanka, Vietnam, Madagascar, and Indonesia. Table 2 provides a valuable insight into the countries' CEI, DC, PV, CLA and ER between the mentioned time periods. Moreover, the percentage values show the changes that have taken place within the variables between the 1st period (1992–2002) and 3rd period (2013–2022).

Considering the growth variations of CEI, PV, DC, CLA and ER in China, it shows that CEI increased approximately by 17% from the 1st to the 3rd period, reflecting that the improvement is firm and yet still cautious. DC rose by 255% reflecting that internal demand is improving probably due to the growing domestic market or industrial use of cinnamon products. The decline in the percentage of output volume accounted for by PV can highlight challenges in transmitting the long-term economic value of cinnamon. This can happen given China's strategic diversion of plant production to crops considered more worthwhile or critical. Therefore, though the value of cinnamon as an agricultural product can be noted, it may not have precedence among China's agriculture development projects. However, there is an observable increase of 35% in land planted for cinnamon, which could indicate China pursuing a proactive approach to enhance production capacity to meet future demand or export prospects. This is unlike ER, which declined by 16%. This introduces another layer of complexity. Although the country's exports may be cheap and competitive in the global market, the essential imported inputs may result in artificially inflated prices. These different trends show that while China is committed to cinnamon

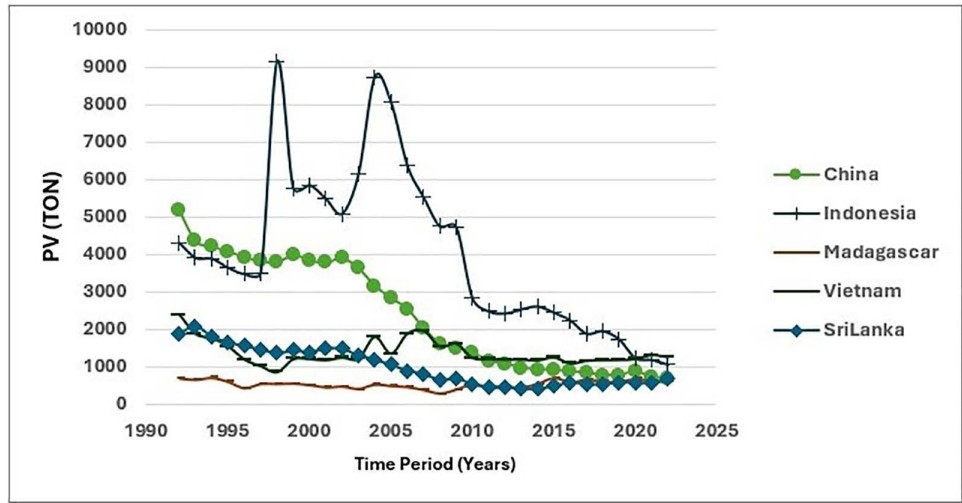

**Fig 10. Fluctuations of PV between 1992-2022.**

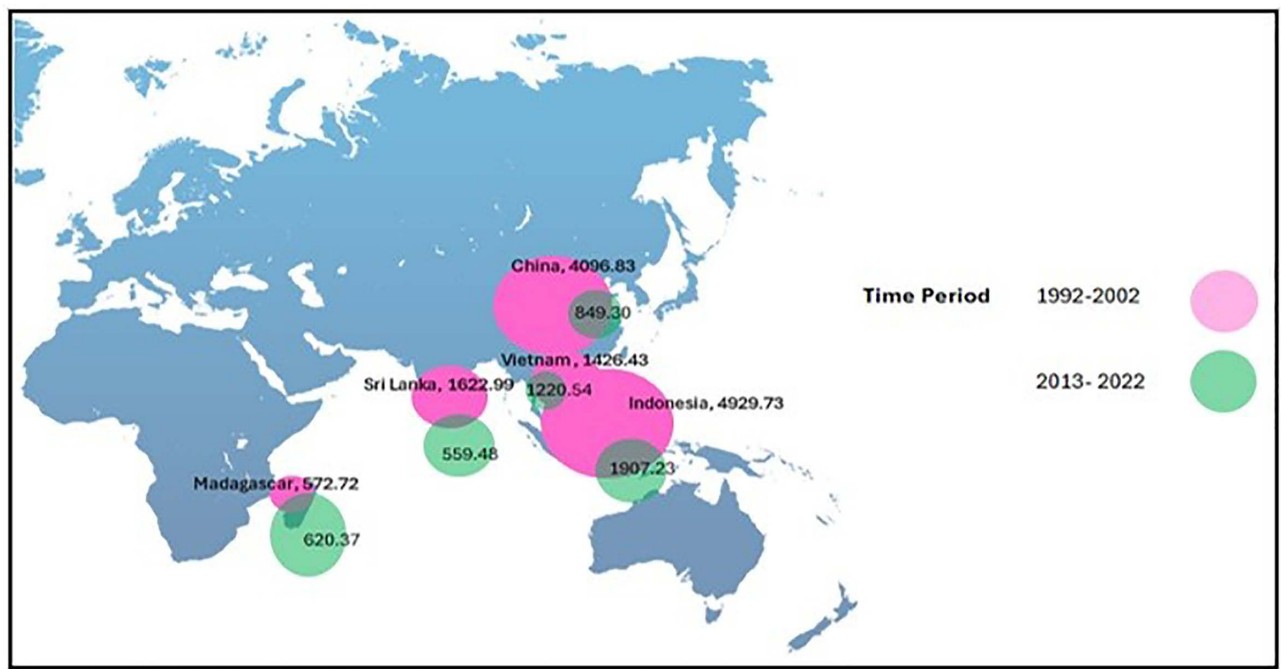

**Fig 11. Bubble map for PV.**

sector development, resource allocation is balanced on a precarious base with market competitiveness and economic constraints.

In addition, the growth variations of Vietnam have been calculated to articulate the trend lines. As a result, it shows that CEI in Vietnam has reached approximately 146% and consolidated the country's position in the global cinnamon market. The huge increase in CEI reflects an upward trend in the leading exporter position of Vietnam. DC, however, has risen only by about 32%, which shows that though the exports are performing well, the internal growth of the product is growing at a slower pace compared to other countries. The slight reduction of PV around −14% only shows that there is stability in the production of cinnamon, implying that the capacity to produce, though not at a high-increasing rate, is maintained.

The most significant change is the considerable expansion of the geographical land allocated for cinnamon cultivation. This sudden increase is indicative of deliberate efforts made by the Vietnamese government to increase production due to the increased and growing world demand. The favourable climatic conditions also enable the use of this increased land, making Vietnam a viable area for cinnamon farming. The ER has been increased by around 78%, which is a signal of currency depreciation. This thus perhaps suggests that while Vietnam's cinnamon exports might, in all respects, enjoy a weak currency to make them cheaper and more competitive worldwide, the economic conditions accompanying devaluation may pose some challenges for the country to manage its costs in keeping with the attainment of economic stability.

Here it shows the growth variations of Sri Lanka. Accordingly, it represents the overall growth of Sri Lanka's CEI as growth near to 39%, reflecting thereby that Sri Lanka's exports of cinnamon have grown at a constant rate throughout the period under consideration. This also reflects the gradual and continuous effort on the part of Sri Lanka to support its lead in the global cinnamon market. Giving support to this is that DC has also recorded growth by 55% nearly, showing a trend upwards in internal demand for cinnamon. The fall in PV of −66% reflects steep falls in production due to unfavourable environmental conditions, labour shortages, and fierce competition in the head of the market. Despite

**Table 2. Average values calculated for CEI, DC, PV, CLA and ER.**

**China**

| Variable | CEI (USD) | DC (Kg) | PV (Ton) | CLA (Ha) | ER (USD) |
|---|---|---|---|---|---|
| **1992-2002** | 0.1934 | 0.0053 | 4096.8331 | 0.0007 | 7.8400 |
| **2003-2012** | 0.1240 | 0.0185 | 2108.2897 | 0.0009 | 7.3653 |
| **2013-2022** | 0.2262 | 0.0189 | 849.3013 | 0.0009 | 6.5581 |
| **% Change between the 1st period and 3rd period** | **16.9745** | **255.4673** | **−79.2693** | **35.4167** | **−16.3507** |

**Vietnam**

| Variable | CEI (USD) | DC (Kg) | PV (Ton) | CLA (Ha) | ER (USD) |
|---|---|---|---|---|---|
| **1992-2002** | 0.2718 | 0.0140 | 1426.4327 | 0.0027 | 12540.596 |
| **2003-2012** | 0.1645 | 0.0509 | 1517.2893 | 0.0121 | 17253.187 |
| **2013-2022** | 0.6691 | 0.0186 | 1220.5383 | 0.0283 | 22337.572 |
| **% Change between the 1st period and 3rd period** | **146.1824** | **32.4762** | **−14.4342** | **950.6735** | **78.1220** |

**Sri Lanka**

| Variable | CEI (USD) | DC (Kg) | PV (Ton) | CLA (Ha) | ER (USD) |
|---|---|---|---|---|---|
| **1992-2002** | 4.6300 | 0.1732 | 1622.9908 | 0.0360 | 64.0200 |
| **2003-2012** | 4.5379 | 0.0870 | 826.7416 | 0.0408 | 108.7262 |
| **2013-2022** | 6.4402 | 0.2677 | 559.4788 | 0.0492 | 157.3754 |
| **% Change between the 1st period and 3rd period** | **39.0984** | **54.5580** | **−65.5279** | **36.5086** | **145.8222** |

**Madagascar**

| Variable | CEI (USD) | DC (Kg) | PV (Ton) | CLA (Ha) | ER (USD) |
|---|---|---|---|---|---|
| **1992-2002** | 0.3821 | 0.0685 | 572.7243 | 0.0010 | 948.6537 |
| **2003-2012** | 0.4388 | 0.0304 | 445.4960 | 0.0011 | 1937.0831 |
| **2013-2022** | 0.5347 | 0.0016 | 620.3739 | 0.0020 | 3367.5434 |
| **% Change between the 1st period and 3rd period** | **39.9479** | **−97.6719** | **8.3198** | **103.6996** | **132.9619** |

**Indonesia**

| Variable | CEI (USD) | DC (Kg) | PV (Ton) | CLA (Ha) | ER (USD) |
|---|---|---|---|---|---|
| **1992-2002** | 0.5874 | 0.0683 | 4929.7259 | 0.0052 | 5421.8774 |
| **2003-2012** | 0.1763 | 0.2270 | 5219.7610 | 0.0083 | 9285.7437 |
| **2013-2022** | 0.3218 | 0.1118 | 1907.2343 | 0.0085 | 13452.983 |
| **% Change between the 1st period and 3rd period** | **−45.2159** | **63.7873** | **−61.3116** | **63.4622** | **148.1240** |

Source: Authors' compilation.

these factors, Sri Lanka has been able to record a growth of around 37% in land areas allocated for cinnamon production and, therefore, has reiterated its commitment to increasing output levels to retain its position as one of the leading exporters of cinnamon.

This reflects an increase in land use and, so, points to the long-term plan of the country in increasing its capacity in view of the external threats. Thirdly, the ER has appreciated by 146%, driven by the fall in the value of the Sri Lankan rupee. This makes the Sri Lankan cinnamon exports competitive at an international level, but on the other side, it delineates general economic difficulties that might affect production costs and have some consequence for the overall stability of the market.

Considering the growth variations of Madagascar, it shows that CEI has increased at a rate of nearly 40% within the selected period, reflecting steady progress in export revenues below the attainment of more significant gains by other countries. In fact, the cinnamon industry in Madagascar is a very interesting one. Considering the DC, the industry has been severely cut back, implying that there is a focus on export rather than the satisfaction of the local demand. The PV increased by 8%, and this will reflect stable production from a different perspective of economic activity, as consumption has been reduced in the economy of Madagascar.

The most evident is the increase in CLA in the use of cinnamon in Madagascar by a factor of approximately 104% due to the attempts of Madagascar to increase the land that the country harvests in order to boost the amount of exportation of that product. This is evident in that the ER increased by a factor of 133% due to the immense depreciation of the national currency. This has increased the efficiency of the exports of Madagascar in the international markets of cinnamon, as well as the fact that Madagascar is faced with greater economic challenges due to this new exchange rate.

Finally counting the growth variations of Indonesia, it depicts that CEI has fallen about −45% because of increasing pressure from other competing exporting countries and a change in the global market circumstances of cinnamon. Though the export revenues have fallen drastically, DC is higher by 64% during the 3rd period compared to the 1st period, representing excellent internal demand and production capacity. While this is the case, PV has shrunk around by −61%, due to pressing environmental challenges and agricultural constraints to overall productivity.

Despite all these developments, Indonesia has expanded the land area utilized in cinnamon production, by 63%, indicative of its efforts toward expanded cultivation perhaps to improve future export performance. Despite the short-term revenue losses in exports, the rise in land use shows a long-term strategy for regaining competitiveness in the global market. Thirdly, the ER has gone up by 148%, showing a noticeable devaluation of the currency which might make exports competitive at an international level but reflect more general economic challenges.

The common trends in the variables ER and DC represent that there is a higher focus on export strategies and difficulties of maintaining a stable domestic market within all the five countries. All the countries, except Indonesia showed an increase in CEI. PV has undergone many fluctuations within all five countries within the respected period.

Meanwhile having a deep understanding of the specific roles that key variables play in different national contexts is a must to identify the determinants of the success of cinnamon exports. To outline the unique strategies each country adopts to maximize their assets, here it has checked the correlations specific to each country between CEI and ER, CLA, PV and DC.

According to the Fig 12, the analysis identified that Sri Lanka as the leading exporter, highlighting especially cinnamon, and displays a strong positive relationship between the CEI and both the CLA and DC. This finding highlights Sri Lanka's significant agricultural and economic commitment to the cultivation of cinnamon, where increased land allocation and increased DC are linked to increased export revenues. Vietnam presents a moderate positive relationship between CLA and CEI, representing some alignment between land utilization and returns on exports, but not as strong as the relationship evident with Sri Lanka. Likewise, Madagascar shows a high relationship between DC and CEI, illustrating domestic consumption and production patterns playing an influential part within export activity. China and Indonesia, with their larger GDP and variable exchange rates, display no strong connection with CEI, indicating the economic significance of cinnamon is limited within both countries. Overall, nations with increased land allocation and internal resources towards cinnamon cultivation (Sri Lanka, with some contribution from Madagascar and Vietnam) have demonstrated higher relative export revenues, highlighting the significance of committed agricultural endeavour over economics as scale as a determining factor within cinnamon export.

Before running the simple linear regression, the stationarity characteristics of the variable were checked through the application of the Levin–Lin–Chu (LLC) test on unit roots. The test was applied on the entire dataset for all countries. The LLC test results are presented in Table 3.

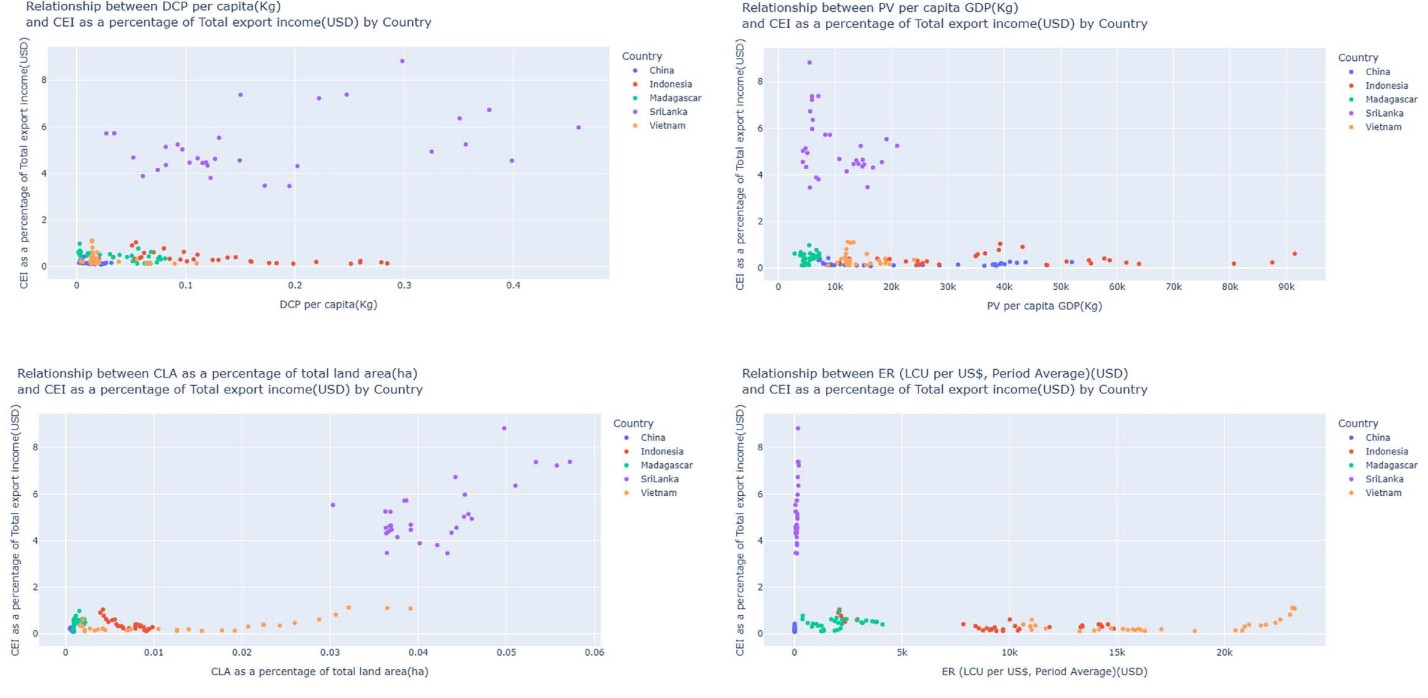

**Fig 12. Country level correlation analysis.**

**Table 3. Average values calculated for CEI, DC, PV, CLA and ER.**

| Variable | 1st Difference Variable | Level LLC | First Difference (LLC) |
|---|---|---|---|
| Log_CEI | D_CEI | −0.44 | −12.03*** |
| Log_DCP | D_DCP | −1.59* | −14.02*** |
| Log_PV | D_PV | −1.25 | −7.84*** |
| Log_CLA | D_CLA | 3.07 | −12.13*** |
| Log_ER | D_ER | 0.34 | −5.75*** |

Note: LLC unit root test – $H_0$: series contains unit roots and $H_1$: series is stationary. The symbols ***, **, * represent 1%, 5%, and 10% significance levels, respectively.

Source: Authors' calculations based on panel data using R 'plm' package.

From the level form analysis, the null hypothesis of non-stationarity could not be rejected for any of the variables namely, Log_CEI, Log_PV, Log_CLA, and Log_ER, at the 5% level of significance, while for Log_DCP, it showed marginal significance at 10% level. This shows that a good portion of the variables exhibit unit root properties in the level form, thus implying non-stationarity. Nevertheless, when the first differencing transformation was applied, all variables D_CEI, D_DCP, D_PV, D_CLA, and D_ER significantly rejected the null hypothesis of a unit root at the 1% significance level. This outcome suggests that the differenced series exhibit stationarity, thereby allowing for the conclusion that all variables can be classified as integrated of order one, I(1).

After checking the staionarity of the variables the simple linear regressions was carried out to analyse the impact of all five variables of CEI.

The four independent variables and the dependent variable in each of the five countries were analysed using SLR, as indicated in the analytical model. The line graphs that follow are based on these findings.

Figs 13 and 14 illustrate how PV impacts CEI. Climatic factors have a significant impact on agricultural yields, which in turn determines export potential [41]. A few of the major factors that have detrimental impacts on production and thus performance of exports are climatic changes, fertilizers usage, and their interaction effects on soil quality [42]. Sri Lanka exhibits the detrimental effects of PV with CEI, and the country's highest $R^2$ value is approximately 12%, which signifies that relatively an adequate share of the PV falls to the income of cinnamon exports. It can be assumed that the growth in volumes of production significantly influences CEI of Sri Lanka and corresponds to the position of this country in the world market of cinnamon and its capabilities to turn efforts into export income, and Sri Lanka reflective negative impact 0.85 Kg for a rise of 1% CEI. Production not meeting demand, damage to reputation, and loss of market share amongst others

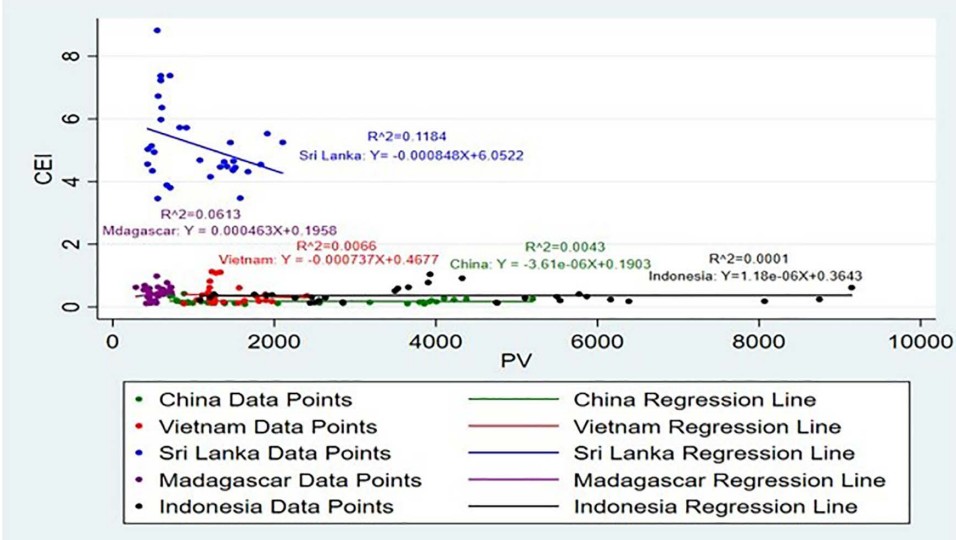

**Fig 13. Scatter Plots for PV vs CEI simple linear regression analysis.**

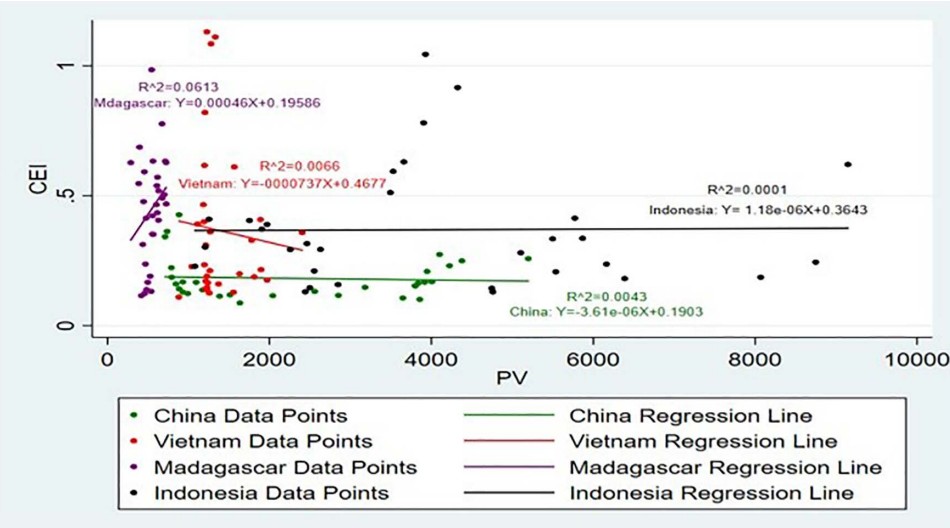

**Fig 14. Scatter plot for PV vs CEI simple linear regression analysis (Without Sri Lanka).**

and all these are considered major risks arising from such problems in production [43]. Indonesia PV demonstrates positive impact and 0.001% define the CEI, which gives lowest adequate PV for CEI.

Notably variations in CEI that is explained by PV changes in Madagascar, Vietnam, and China are 6%, 0.06%, and 0.04%, respectively. Madagascar PV has a positive impact with CEI and other two countries have negative impact. This again proves the very limited impact of PV on CEI. The cinnamon plant requires a climate characterized by high humidity and warmth combined with an equally fertile type of soil. The production of seaweed positively impacts exports in low- and middle-income countries by driving economic growth and diversifying export portfolios [44]. This would mean that the PV alone does not have a huge effect on these countries' CEI. In Madagascar Growth of 1% of CEI brings an enormous increase approximately by 0.46 ton in PV. Enhancing production efficiency can lead to a substantial increase in products, enabling Ethiopia to significantly elevate its export volumes and strengthen its position in the global market [45]. Production at present, however, indicates that none of the above ideal conditions is increasingly hard to sustain.

Figs 15 and 16 show the impact of DC on CEI. which supports prior research that strong domestic demand improves production effectiveness and global competitiveness [33]. Accordingly, it is noticeable that only the DC Sri Lanka has a positive impact on CEI, while the ER China, Madagascar, Indonesia and Vietnam have a negative impact on CEI. DC plays a crucial role in driving the exportation of agricultural commodities. High-income countries like EU28 are self-sufficient because most of the primary crops rely on their respective domestic production, it is evident that DC has a positive impact on cinnamon exports [46]. In the case of Vietnam, Madagascar, Indonesia and China presents a different picture where increase in DC has a negative effect on export availability as has been observed in the case of Saudi Arabia's wheat market implying that increase in domestic consumption may reduce exportable surplus [47]. This thus implies that there could be other influencing variables on the CEI for Sri Lanka and Vietnam and more explore is necessitated to identify what other factors influence export earnings in these countries. High DC can limit exports by reducing the available supply for international markets and increasing the opportunity costs associated with the usage of products [48].

The highest $R^2$ values are for Madagascar and Indonesia, which is 40%. This shows that 40% of the independent variable DC is explained by the dependent variable CEI and this SLR model is a suitable fit for the analysis in these two countries. The $R^2$ values of Sri Lanka and Vietnam, which is 14% and 11% respectively, shows that SLR is slightly suitable

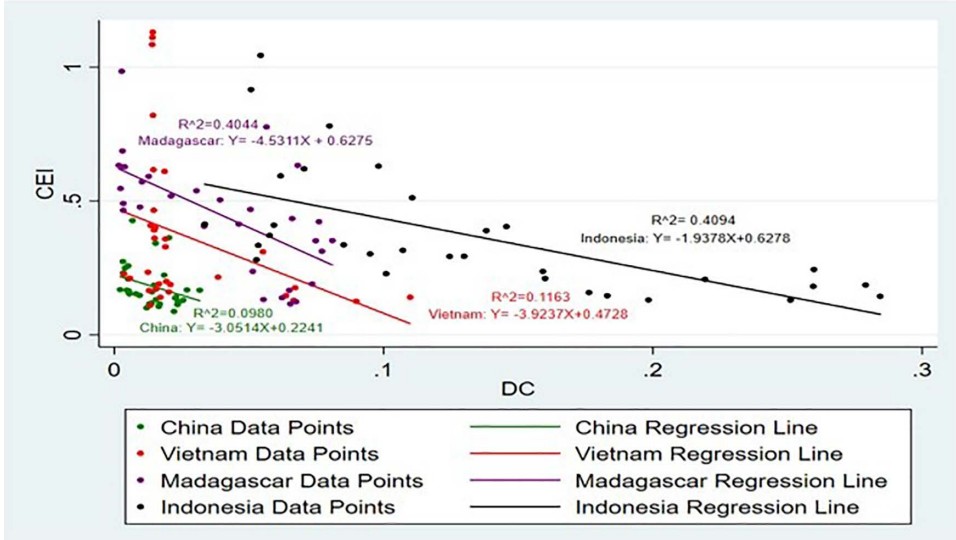

**Fig 15. Scatter Plot for DC and CEI simple linear regression analysis.**

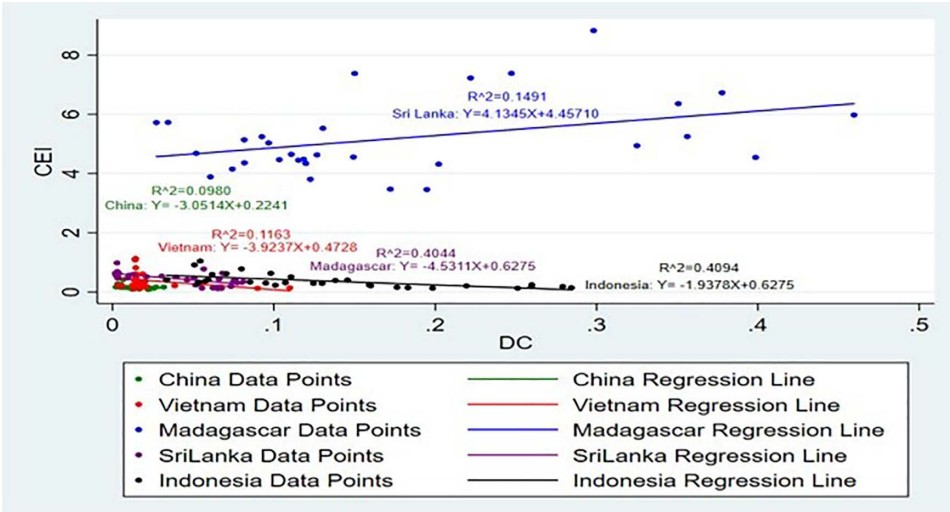

**Fig 16. Scatter Plot for DC and CEI simple linear regression analysis (Without Sri Lanka).**

for this model. The lowest model fit shows by China, it is approximately 10%. These results make it easier to understand the complexity of the influence of DC on CEI in the light of country specific economic contexts and highlighting country specific factors driving export dynamics.

Figs 17 and 18 show the impact on CLA on CEI. Land area suitability also influences production [49]. The positive correlation between CLA and CEI in all the analysed countries highlights the need to increase the area of land used for cinnamon production [50,51]. This finding implies that expansion of the area that is allocated for cinnamon can lead to a direct improvement in the production capacity and therefore export volumes, supporting the call for efficient land use and agricultural intensification [52]. Accordingly, it is noticeable that CLA has a positive and negative impact on CEI.

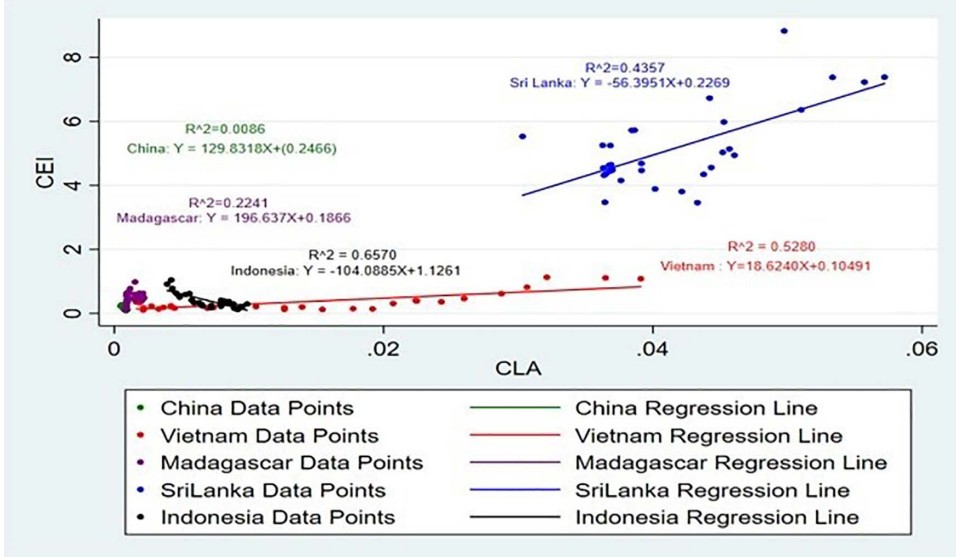

**Fig 17. Scatter Plot for CLA and CEI simple linear regression analysis.**

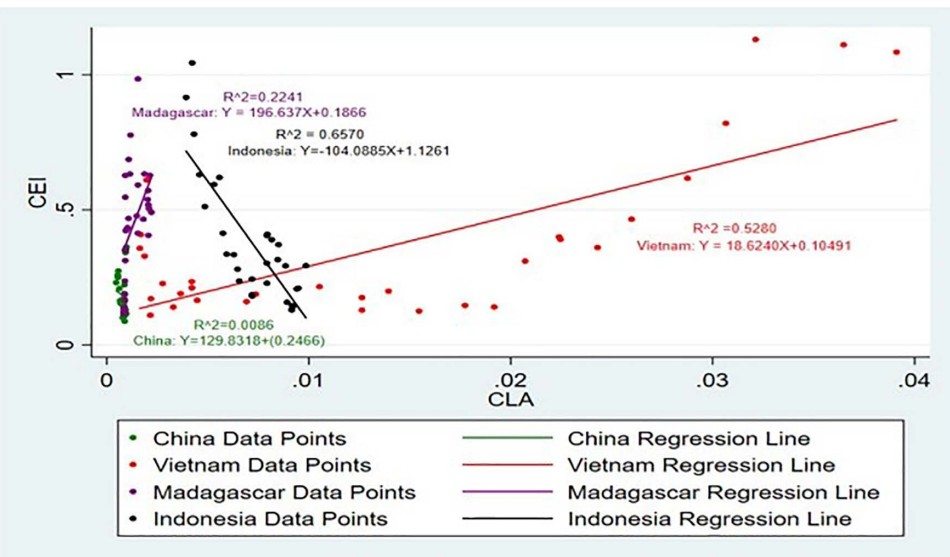

**Fig 18. Scatter Plot for CLA and CEI simple linear regression analysis (Without Sri Lanka).**

Seasonal weather forecasting comes into play in terms of the optimization of fertilizer application to maintain consistent yields of crops. Such precision helps boost not only agricultural productivity but also export performance through a stable supply of maize both in domestic and international markets [53]. CLA of Vietnam, Madagascar and China has a positive impact on CEI and growth of 1% of CEI brings an enormous increase approximately by 19 ha, 197 ha and 130 ha, while Sri Lanka and Indonesia has a negative impact on CEI and enhance of 1% of CEI brings an enormous decrease roughly by 56 Ha and 104 Ha. Regarding the agricultural land area, an upward impact on the exports, especially amidst the high international demand for agricultural commodities [51] will be brought about by an increase in land set aside for agriculture. The land area significantly impacts the output exports, as agricultural and residential areas generate higher concentrations of pollutants in runoff compared to forested areas [54].

Moreover, when the $R^2$ values are taken into consideration, the highest $R^2$ value is from Indonesia, which is 65%. It suggests that the SLR model is a highly suitable and strong predictor for this analysis. And the $R^2$ values of Vietnam and Sri Lanka are 52% and 43% respectively, which shows that SLR is suitable for this analysis.

Figs 19 and 20 represent the impact of ER on CEI. A favourable ER significantly increases export competitiveness [55]. Accordingly, it shows that the ER of Sri Lanka, Vietnam and Madagascar has a positive impact on CEI, while the ER of Indonesia and China has a negative impact on CEI. Higher competitiveness in goods relative to related goods from other countries, whose exchange rates are lower, can be pushed down. In this way, the lower demand induced by higher exchange rates can make firms decrease the export price to stay competitive [56]. This has been particularly seen in the case of Turkey, where export growth has been observed to strongly relate with the changes in ER [57]. The decrease in the exchange rate value of the national currency US dollar emerges as a favourable condition that influences export competitiveness increase for the agriculture industry of the developing countries [58]. Moreover, when the $R^2$ values are taken into consideration it is noticeable that Indonesia and Sri Lanka have the highest $R^2$ value of 0.3939 and 0.4327. It shows that 40% and 43% of the independent variable, ER is explained by the dependent variable, CEI respectively.

In addition, it shows that when CEI is increased by 1% in the countries Sri Lanka and China ER is also increased by 0.0195 USD and 0.0179 USD respectively. This positive impact on the stochastic value suggests that, for these three countries, export income growth may be associated with stability or strength in the currency. An effective ER may

**Fig 19. Scatter plot for ER and CEI simple linear regression analysis.**

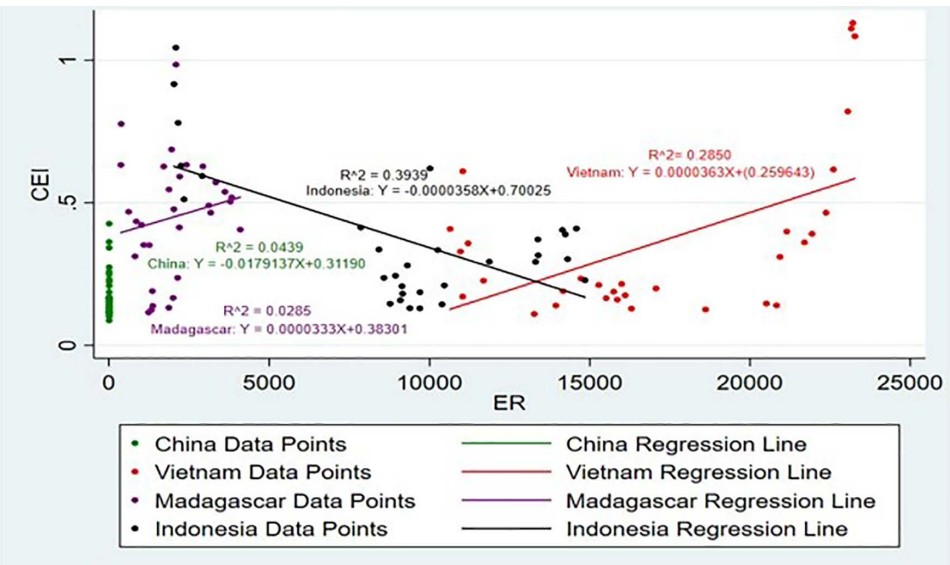

**Fig 20. Scatter plot for ER and CEI simple linear regression analysis (Without Sri Lanka).**

contribute to the exportation business, trade policies, market access, and cost of production have also been key critical factors in this aspect [59]. Also, it is noticeable that when CEI is increased by 1% in the countries Indonesia and Madagascar the ER has slight change. This would then mean that the negative correlation points to the fact that such countries face downward pressure on their currencies ER can impact positive and negative, ER volatility increases, and the mean performance of exports tends to decline, particularly in the short run [60]. The positive coefficient of the ER in the general model given that a weakened local currency increases export earnings as it cuts down the prices at which the export goods are sold to foreign clients.

The heterogeneity in how PV, DC, ER CLA, and CEI interact across top cinnamon-producing countries reflects divergent structural, policy, and market contexts. In Vietnam, PV negatively impacts CEI primarily because output increases during global price volatility yield diminishing export revenues. This is explained by exchange rate volatility, which long-term suppresses exports. In addition, [61] show a 1% rise in ER volatility reduces export volume by approximately 0.11% and by expanding DC that diverts cinnamon from export channels. In China, similar dynamics apply rising domestic demand and currency strength can offset the gains from higher PV, leading to diminished CEI. Contrastingly, Indonesia leverages PV and ER positively because depreciation enhances competitiveness, and its production increases translate effectively into CEI due to strong export linkages and government support [62,63]. In Sri Lanka, smallholder systems mean PV often undermines CEI, as increased CLA yields may not overcome price shocks and high DC, and occasional ER overvaluation may further erode export profitability [64]. Meanwhile, in Madagascar, PV and ER boost CEI when integrated with export-oriented systems, and CLA expansion efficiently scales exports due to available land and agro-ecological suitability [65]. These divergent findings underscore those variables like PV, ER, and CLA do not act uniformly; their net impacts depend on each nation's trade policy, market structure, and scale capacities.

In conclusion, the results and discussion section prove that the authors of this research have achieved all the objectives. Therefore, it is concluded that PV, DC, ER and CLA has an impact on CEI in the main five cinnamon exporting countries. This study delineates the complex interactions between various economic and environmental variables and their collective impact on the cinnamon export industry. It highlights the necessity for targeted policies that foster sustainable agricultural practices, support economic stability, and enhance market responsiveness to changes in domestic and global economic conditions. Future research should continue to explore these variables in evolving global contexts to better understand their long-term impact of agricultural exports.

## 4. Conclusion and recommendations

The current study aimed at determining the role of PV, DC, ER, and CLA in shaping CEI in the following major export countries: China, Sri Lanka, Vietnam, Indonesia, and Madagascar. The findings have shown that these factors have denoted both positive and negative impacts affecting CEI in different circumstances. For example, in both Vietnamese and Chinese export sectors, PV and DC act as challenges to be addressed by optimizing export facility management and stabilizing exchange rates. In Indonesia's case, it showcases favourable ER and PV but must address CLA's negative effect by emphasizing yields over land area increases. In Sri Lanka's export performance, it must mitigate CLA's negative effect by effectively managing land and prices. Madagascar's results showed positive effects; thus, it should improve CEI by sustainably using land and by developing export facilities. In general, all countries require strategies focusing on rationalized production planning and management, land management strategies, and agricultural educational and capacity development efforts. All these steps would ensure increased CEI performance to be competitive in the global market. To generalize results to be relevant to all export countries, it would be advisable to have all countries part of new studies in the future. Additional variables to be examined should target climate changes and quality issues associated with export quality.

## Supporting information

**S1 Appendix. Data file.**
(XLSX)

## Author contributions

**Conceptualization:** Krishantha Wisenthige, Ruwan Jayathilaka, Umesha Dabare, Thisalya Marasinghe, Malki Radeesha, Nethmi Kavindya.

**Data curation:** Ruwan Jayathilaka, Thisalya Marasinghe, Malki Radeesha.

**Formal analysis:** Ruwan Jayathilaka, Umesha Dabare, Thisalya Marasinghe, Malki Radeesha.

**Investigation:** Krishantha Wisenthige, Ruwan Jayathilaka, Fiona Ann.

**Methodology:** Krishantha Wisenthige, Ruwan Jayathilaka, Nethmi Kavindya.

**Project administration:** Krishantha Wisenthige.

**Supervision:** Krishantha Wisenthige.

**Validation:** Krishantha Wisenthige, Ruwan Jayathilaka, Fiona Ann.

**Visualization:** Malki Radeesha, Nethmi Kavindya.

**Writing – original draft:** Umesha Dabare, Thisalya Marasinghe, Malki Radeesha, Fiona Ann, Nethmi Kavindya.

**Writing – review & editing:** Krishantha Wisenthige, Ruwan Jayathilaka, Umesha Dabare, Thisalya Marasinghe, Fiona Ann, Nethmi Kavindya.

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
