## [Decision Letter · Decision Letter 0]

2 May 2025

Dear Dr. Wisenthige,

Thank you for submitting your manuscript to PLOS ONE. After careful consideration, we feel that it has merit but does not fully meet PLOS ONE’s publication criteria as it currently stands. Therefore, we invite you to submit a revised version of the manuscript that addresses the points raised during the review process.

We look forward to receiving your revised manuscript.

Kind regards,

S Ezhil Vendan, Ph.D

Academic Editor

PLOS ONE

Additional Editor Comments:

Please check the manuscript structure with respect to the journal guidelines. The introduction section is too long, please reduce the general statements and strict to the rationale of the study objectives. Avoid sub headings in the introduction section. Based on the reviewers comments and raised queries, the manuscript needs major revision.

Reviewers' comments:

Reviewer's Responses to Questions

**Comments to the Author**

1. Is the manuscript technically sound, and do the data support the conclusions?

Reviewer #1: Partly

Reviewer #2: Partly

2. Has the statistical analysis been performed appropriately and rigorously?

Reviewer #1: No

Reviewer #2: No

3. Have the authors made all data underlying the findings in their manuscript fully available?

Reviewer #1: Yes

Reviewer #2: Yes

4. Is the manuscript presented in an intelligible fashion and written in standard English?

Reviewer #1: Yes

Reviewer #2: Yes

Reviewer #1: The manuscript, "Key Determinants of Cinnamon Export Income: Insights from the World's Top Five Producers", explores the factors influencing cinnamon export income (CEI) for five major exporting countries (China, Sri Lanka, Indonesia, Madagascar, and Vietnam) over three decades. While the study provides valuable insights and has strong potential, it also has certain limitations that could be addressed to strengthen its contributions. Below is a critical review with suggestions for improvement and examples:

Strengths

1. Timely and Relevant Topic: The focus on cinnamon export income is pertinent, given the spice's global demand and economic significance. The emphasis on analyzing multiple countries adds depth to the investigation.

2. Well-Defined Objectives: The study clearly outlines primary and secondary objectives, aligning them with the research gap identified in existing literature.

3. Data Coverage: The analysis spans three decades (1992–2022), providing a robust time series for examining trends and relationships.

4. Multifactor Analysis: By incorporating production volume (PV), domestic consumption (DC), exchange rate (ER), and cultivated land area (CLA), the study provides a multidimensional view of the factors affecting CEI.

Limitations and Suggestions for Improvement

1. Theoretical Framework

o Limitation: The theoretical framework connecting the identified factors to CEI is underdeveloped. While the study identifies PV, DC, ER, and CLA as key variables, it does not adequately explain the mechanisms through which these factors interact or influence CEI.

o Suggestion: Introduce a conceptual model or flowchart showing causal relationships and interdependencies among the variables. For instance, explain how exchange rate volatility impacts both price competitiveness and market stability.

2. Methodological Justification

o Limitation: The choice of a simple linear regression model may oversimplify the complex relationships among the variables. Factors like exchange rate and domestic consumption could have nonlinear impacts on CEI.

o Suggestion: Consider adopting more advanced econometric models, such as panel data regression or structural equation modeling, to capture nuanced relationships and address potential multicollinearity.

o Example: For exchange rates, explore whether their impact varies with changes in market conditions (e.g., fluctuations in global demand).

3. Inconsistent Findings

o Limitation: The study reports conflicting impacts of PV and CLA on CEI across countries (e.g., PV has a negative impact in some countries but a positive impact in others). While these findings are interesting, the manuscript does not explore the underlying reasons for these differences.

o Suggestion: Add a discussion on contextual factors (e.g., differences in trade policies, agricultural practices, or market structures) that could explain these variations.

o Example: Investigate whether Indonesia's positive PV-CEI relationship is influenced by government subsidies or trade incentives.

4. Limited Analysis of Policy Implications

o Limitation: The study briefly mentions implications for policymakers, but these are not detailed or actionable. For instance, recommendations for managing exchange rate volatility are lacking.

o Suggestion: Provide concrete policy recommendations tailored to each country, focusing on areas such as foreign exchange risk management, technological innovation in agriculture, and export promotion strategies.

5. Data Presentation

o Limitation: While the manuscript includes some descriptive statistics, the data presentation is limited in terms of visual appeal and explanatory detail.

o Suggestion: Add more graphs, charts, or tables to visualize key findings. For example, use a comparative bar graph to show country-level differences in the impact of DC or CLA on CEI.

o Example: A scatter plot illustrating the relationship between ER fluctuations and CEI across the study period would enhance reader comprehension.

6. Narrow Scope of Study

o Limitation: The study focuses exclusively on cinnamon. While this adds specificity, the findings may not generalize to other spices or agricultural exports.

o Suggestion: Acknowledge this limitation and propose future research extending the analysis to other high-value crops or spices to validate the generalizability of findings.

7. Literature Review

o Limitation: The literature review does not adequately compare the current study's contributions to prior research, nor does it highlight its novelty.

o Suggestion: Expand the literature review to include a critical assessment of gaps in prior studies and explicitly state how the current research fills these gaps.

8. Language and Formatting

o Limitation: The manuscript contains instances of awkward phrasing and grammatical errors, which may detract from its professional presentation.

o Suggestion: Perform thorough proofreading and editing to improve clarity and coherence.

Examples of Specific Improvements

• Clarity in Data Interpretation: Discuss why DC negatively impacts CEI in most countries but positively in Sri Lanka. Hypothesize that higher domestic consumption in Sri Lanka indicates robust internal demand, stabilizing export prices.

• Improved Visualization: Add a line graph showing annual CEI trends for each country, overlaying global cinnamon price indices for context.

• Policy Insights: Suggest that Sri Lanka adopt integrated marketing campaigns to leverage its unique position as a supplier of True Cinnamon.

By addressing these limitations and implementing the suggested improvements, the manuscript can significantly enhance its academic rigor, practical relevance, and overall impact.

Reviewer #2: Dear authors

You wrote a good article for the improvement of agricultural international trade. Unfortunately, you could not show the novelty of this study well. In addition, you used the SLR method, while secondary data has non-stationary data that cause spurious regression. This issue cannot be solved with SLR but must be solved with advanced econometric models that adopt unit root tests. For that, you need to consider advanced econometric models such as ARDL, ECM or others.

Best wishes

**Do you want your identity to be public for this peer review?** For information about this choice, including consent withdrawal, please see our Privacy Policy

Reviewer #1: **Yes: ** Md Takibur Rahaman, PhD

Reviewer #2: **Yes: ** Agus Dwi Nugroho

---

## [Author Response · Author response to Decision Letter 1]

3 Jul 2025

Please refer to the response to reviewers document uploaded.

---

## [Decision Letter · Decision Letter 1]

27 Jul 2025

Dear Dr. Krishantha Wisenthige,

Thank you for submitting your manuscript to PLOS ONE. After careful consideration, we feel that it has merit but does not fully meet PLOS ONE’s publication criteria as it currently stands. Therefore, we invite you to submit a revised version of the manuscript that addresses the points raised during the review process.

**ACADEMIC EDITOR:**

We look forward to receiving your revised manuscript.

Kind regards,

S Ezhil Vendan, Ph.D

Academic Editor

PLOS ONE

Journal Requirements:

Additional Editor Comments:

The revised manuscript is not satisfactory. Authors have not responded to the earlier (editor’s) comments. Still, the manuscript needs major revision with respect to the following comments;

Authors should check the manuscript organization/structure with respect to the journal guidelines (Manuscript organization: Title, Abstract, Introduction, Materials and Methods, Results, Discussion, and Conclusions) and revise the manuscript.

The introduction section is too long, please reduce the general statements and strict to the rationale of the study objectives.

Authors should avoid sub headings in the introduction section and should reduce the introduction section.

Reviewers' comments:

Reviewer's Responses to Questions

**Comments to the Author**

Reviewer #1: All comments have been addressed

Reviewer #2: (No Response)

2. Is the manuscript technically sound, and do the data support the conclusions?

Reviewer #1: Yes

Reviewer #2: No

3. Has the statistical analysis been performed appropriately and rigorously?

Reviewer #1: Yes

Reviewer #2: No

4. Have the authors made all data underlying the findings in their manuscript fully available?

Reviewer #1: Yes

Reviewer #2: Yes

5. Is the manuscript presented in an intelligible fashion and written in standard English?

Reviewer #1: Yes

Reviewer #2: (No Response)

Reviewer #1: I am happy that the authors considered my concern seriously and successfully addressed all the comments. Can be accepted, i don't have any further comments

Reviewer #2: Dear authors

This article is very good but unfortunately you missed the unit root test which is mandatory for secondary data.

Best wishes

**Do you want your identity to be public for this peer review?** For information about this choice, including consent withdrawal, please see our Privacy Policy

Reviewer #1: **Yes: ** Dr. Md. Takibur Rahaman

Reviewer #2: **Yes: ** Agus Dwi Nugroho

---

## [Author Response · Author response to Decision Letter 2]

11 Sep 2025

Kindly refer the response to the reviewer document for the author's response to reviewer and editor comments.

---

## [Editor Report · Decision Letter 2]

22 Sep 2025

Dear Dr.  Wisenthige,

Thank you for submitting your manuscript to PLOS ONE. After careful consideration, we feel that it has merit but does not fully meet PLOS ONE’s publication criteria as it currently stands. Therefore, we invite you to submit a revised version of the manuscript that addresses the points raised during the review process.

We look forward to receiving your revised manuscript.

Kind regards,

S Ezhil Vendan, Ph.D

Academic Editor

PLOS ONE

Journal Requirements:

Additional Editor Comments:

The revised manuscript is not satisfactory with respect to the manuscript organization. As per the journal guideline, research article manuscript should be organized with the content of abstract, introduction, materials and methods, results and discussion, and conclusions (optional). Please check the manuscript structure and authors should revise the manuscript with respect to the journal guidelines for manuscript organization.

As research article, remove elaborated literature review with the subheading “Literature review” in the introduction section.

Check the subheadings “Data and methodology”, “Results and Discussion” and “Conclusion and recommendations” and write appropriate subheadings and revise the manuscript as per the journal guidelines for manuscript organization.

Please check errors in scientific name of plants (italic, capital & small letter fonts) throughout the manuscript (e.g., Line 116) and revise.

---

## [Author Response · Author response to Decision Letter 3]

9 Nov 2025

Please refer the response to the reviewers' document attached.

---

## [Editor Report · Decision Letter 3]

19 Nov 2025

Dear Dr. Wisenthige,

**The manuscript needs minor revision.**

We look forward to receiving your revised manuscript.

Kind regards,

S Ezhil Vendan, Ph.D

Academic Editor

PLOS ONE

**Journal Requirements:**

**Additional Editor Comments:**

The manuscript is improved compared to the previous version. Still, the manuscript needs revision for considerable further. Please revise the with respect to the following comments;

Line 123-124: Write expansion of PV, CLA, ER and DC.

Line 129: Write expansion of CEI.

The conclusion section is too elaborative. Should brief the conclusions with respect to the highlight of the obtained results and write recommendations in few lines only.

---

## [Author Response · Author response to Decision Letter 4]

24 Nov 2025

Please refer the uploaded author responce document

---

## [Editor Report · Decision Letter 4]

25 Nov 2025

Unlocking Cinnamon Export Success: Key Determinants from the World's Top Five Producers

PONE-D-25-10270R4

Dear Dr. Krishantha Wisenthige,

We’re pleased to inform you that your manuscript has been judged scientifically suitable for publication and will be formally accepted for publication once it meets all outstanding technical requirements.

Kind regards,

S Ezhil Vendan, Ph.D

Academic Editor

PLOS ONE
---

## [Editor Report · Acceptance letter]

PONE-D-25-10270R4

PLOS ONE

Dear Dr. Wisenthige,

I'm pleased to inform you that your manuscript has been deemed suitable for publication in PLOS ONE. Congratulations! Your manuscript is now being handed over to our production team.

Kind regards,

on behalf of

Dr. S Ezhil Vendan

Academic Editor

PLOS ONE